# Twenty-Year Retrospective Study of Post-Enucleation Chemotherapy in High-Risk Patients with Unilateral Retinoblastoma

**DOI:** 10.3390/children9121983

**Published:** 2022-12-16

**Authors:** Yoon Sunwoo, Jung Yoon Choi, Hyun Jin Park, Bo Kyung Kim, Kyung Taek Hong, Sang In Khwarg, Jaemoon Koh, Sung-Hye Park, Dong Hyun Jo, Jeong Hun Kim, Jung-Eun Cheon, Hyoung Jin Kang

**Affiliations:** 1Department of Pediatrics, Seoul National University College of Medicine, Seoul 03080, Republic of Korea; 2Cancer Research Institute, Seoul National University, Seoul 03080, Republic of Korea; 3Department of Opthalmology, Seoul National University College of Medicine, Seoul 03080, Republic of Korea; 4Department of Pathology, Seoul National University College of Medicine, Seoul 03080, Republic of Korea; 5Department of Anatomy and Cell Biology, Seoul National University College of Medicine, Seoul 03080, Republic of Korea; 6Department of Radiology, Seoul National University College of Medicine, Seoul 03080, Republic of Korea; 7Wide River Institute of Immunology, Hongcheon 25159, Republic of Korea

**Keywords:** retinoblastoma, post-enucleation, adjuvant chemotherapy, unilateral, high risk factors

## Abstract

Primary enucleation is a life-saving treatment for advanced intraocular retinoblastoma, particularly in patients with poor visual potential and functional contralateral eyes. This single-center study presents the treatment outcomes of patients with unilateral retinoblastoma who received primary enucleation and adjuvant chemotherapy with cyclophosphamide, vincristine, doxorubicin, and intrathecal methotrexate (CVDM) between 2000 and 2020. Twenty patients were enrolled in the study. The median age at diagnosis was 26 months (range, 1–45). Eighteen patients (90%) were in group E and two (10%) were in group D, according to the intraocular classification of retinoblastoma guidelines. Excluding one patient with an inadequate specimen, 19 patients (95%) had optic nerve involvement (ONI) at least up to the lamina cribrosa. Eight patients (40%) had choroidal invasion in addition to ONI. Two patients (10%) were surgical resection margin positive. The overall and event-free survival rates were 100% and 95%, respectively, for a median follow-up duration of 102.24 months (range 24.2–202.9). There were no relapses or deaths due to any cause, but one patient developed secondary rhabdomyosarcoma 99.6 months after chemotherapy. Treatment was well tolerated, with minimal hematotoxicity and hepatotoxicity. CVDM as a post-enucleation chemotherapy for advanced intraocular retinoblastoma has excellent outcomes with tolerable toxicity. However, in line with updated treatment trends, further risk stratification and lowering the treatment intensity should be considered. Continued long-term follow-up is required to further determine late effects.

## 1. Introduction

Retinoblastoma is the most common primary intraocular neoplasm in children, with a known incidence of 1/17,000–20,000 [1,2]. It is caused by mutational inactivation of both alleles of *RB1,* a known tumor suppressor gene [3]. Unilateral retinoblastoma is responsible for up to 70% of the cases [4,5]. The primary goal of retinoblastoma treatment is to improve survival and the secondary goals are to salvage the eye and vision. Primary enucleation has not been as commonly utilized recently with the development of new treatment modalities, including systemic, intra-arterial, and intra-vitreal chemotherapy [6]. However, enucleation remains a life-saving treatment, especially for advanced intraocular disease with poor potential for vision and functional contralateral eye [7]. The histological presence of high-risk factors (HRFs) is important for predicting local recurrence, distant metastasis, and overall survival, dictating further treatment [8,9]. Although large institutions adopt similar protocols, there is no universal international consensus on the appropriate subject, intensity, or duration of adjuvant chemotherapy following enucleated advanced unilateral retinoblastoma [10,11]. A regimen including cyclophosphamide, vincristine, doxorubicin, and intrathecal methotrexate (CVDM) has been in place for post-enucleation chemotherapy in Seoul National University Children’s Hospital since the 1990s [12].

In this retrospective, non-comparative study, we present the treatment outcomes of patients with high-risk unilateral retinoblastoma who underwent primary enucleation followed by adjuvant chemotherapy with CVDM at the Seoul National University Children’s Hospital from 2000 to 2020.

## 2. Patients and Methods

### 2.1. Patients and Study Design

We retrospectively analyzed the electronic medical records of all patients diagnosed with unilateral retinoblastoma who were treated with upfront enucleation followed by chemotherapy at the Seoul National University Children’s Hospital between January 2000 and January 2020. Patients with any prior diagnosis of neoplasm and patients who were transferred and/or lost to follow-up during treatment were excluded. Participants were enrolled after approval from the ethics review board of our institution in accordance with the Declaration of Helsinki.

Patients were diagnosed and staged according to the Reese–Ellsworth (R-E) classification and the intraocular classification of retinoblastoma (ICRB). Demographic data collected included sex and age at diagnosis. The results of *RB1* mutation analysis using polymerase chain reaction, direct sequencing of exons 1–27, multiplex ligation-dependent probe amplification, and fluorescent in situ hybridization were collected when available. All patients underwent initial brain magnetic resonance imaging (MRI) at diagnosis, and spinal MRI was performed when central nervous system (CNS) extension was suspected based on brain MRI findings. All patients underwent routine cerebrospinal fluid (CSF) and bone marrow examinations. Clinical findings were documented using indirect ophthalmoscopy. Histopathologic features of enucleated eyes—including size, growth pattern, retinal involvement, choroidal invasion, degree of optic nerve invasion, extraocular extension, and involvement of the surgical resection margin—were reviewed and classified.

### 2.2. Treatment

All patients were treated with outpatient-based adjuvant chemotherapy with CVDM (Figure 1). Intravenous administration of cyclophosphamide (20 mg/kg), vincristine (0.05 mg/kg), and doxorubicin (2 mg/kg) was administered every 3 weeks until week 21, followed by cyclophosphamide (30 mg/kg) and vincristine (0.05 mg/kg) every 3 weeks until week 57. Intrathecal methotrexate was administered weekly for the first 6 weeks. The dosage and the number of cycles were recorded for each patient.

### 2.3. Toxicity and Response Evaluation

Adverse events were recorded according to the Common Terminology Criteria for Adverse Events (CTCAE) version 4.0. Response evaluation included history and physical examination, repeated ophthalmologic examination under anesthesia, and MRI scans every 3–6 months in the first 2 years. The duration of follow-up and final systemic outcome (relapse, secondary malignancy, and death due to any cause) were recorded. Echocardiography was performed to monitor anthracycline-induced cardiotoxicity. Serial ocular examinations and the development of secondary malignancies were reviewed for late effects.

### 2.4. Statistical Analysis

Statistical analyses were performed using IBM SPSS Statistics for Windows (version 25.0; IBM Corp., Armonk, NY, USA). Continuous parameters are represented by either mean and standard deviation or median and range, depending on the distribution pattern. Categorical parameters are represented by numbers and proportions. Overall survival (OS) and event-free survival (EFS) were calculated and presented using Kaplan–Meier analysis.

## 3. Results

### 3.1. Baseline Characteristics

The baseline characteristics of the patients are presented in Table 1. A total of 20 patients were enrolled: 13 (65%) female and the remaining 7 (35%) male. The median age at diagnosis was 26 months (range: 1–45 months). Eight eyes were on the left and 12 eyes were on the right. Nineteen patients (95%) were in group V and one (5%) was in group IV according to the RE classification, 18 patients (90%) were in group E, and two (10%) were in group D according to the ICRB guidelines.

Histopathological HRFs are presented in Table 2. Except for one patient whose optic nerve was uncheckable due to poor surgical sample and did not have any other HRFs, all 19 patients (95%) had optic nerve involvement (ONI), at least up to the lamina cribrosa. In 3 (15%), the patients’ invasion was intra-laminar, and in 16 (80%) the patients’ invasion extended posterior to the lamina cribrosa (PLONI). Of the three intra-laminar cases, one had isolated intra-laminar ONI and the other two had concomitant choroidal invasion without other HRFs. Of the 16 PLONI cases, 9 (56%) had isolated PLONI, 5 (31%) had concomitant choroidal invasion without any other HRFs, 1 (6%) had three HRFs including choroidal invasion and positive surgical margin, and 1 (6%) had a total of four HRFs including anterior chamber invasion, scleral invasion, and positive surgical margin.

*RB1* gene mutation analysis was performed in 12 (60%) patients, and only 1 (5%) showed *RB1* deletion. Initial brain MRI was performed in all patients, and spinal MRI was performed in four patients. Among them, one (5%) showed leptomeningeal enhancement on brain and spine MRI, suggesting possible CNS invasion. However, CSF cytology of all 20 patients (100%) showed no evidence of malignancy.

### 3.2. Response and Survival

All except one patient underwent a full course of chemotherapy for 57 weeks. In one patient with isolated PLONI, chemotherapy was discontinued at week 46 because of parental withdrawal of consent. All patients had at least 2 years of follow-up. The overall and event-free survival rates were 100% and 95%, respectively, with a median follow-up duration of 102.24 months (range, 24.2–202.9) (Figure 2). No relapse or distant metastasis was observed. There was one second malignancy 99.6 months after chemotherapy. No deaths occurred due to any cause.

### 3.3. Toxicity

Severity of adverse events in chemotherapy are described in grades in the CTCAE. Any events surpassing grade 3—which are severe or medically significant events likely requiring hospitalization—were reported. Twelve (60%) of the patients experienced grade 3 toxicity. Febrile neutropenia, defined as a combination of fever over 38 °C and absolute neutrophil counts of less than 1000, occurred in 11 cases (55%) for a median of 1 day (range 1–5). None had higher risk complications of febrile neutropenia such as proven bacteremia or septic shock. In only three instances (15%) were these the cause of treatment delay. Three cases (15%) showed alanine transferase elevation above 200 mg/dL, representing impaired hepatic function, for 3–24 days. There was no significant decline in renal or cardiac function, as approximated by changes in creatinine clearance, left ventricular dimensions, and left ventricular ejection fraction.

### 3.4. Late Effects

One patient developed a secondary rhabdomyosarcoma of the upper right eyelid 99.6 months after chemotherapy. The tumor size was small and almost undetectable after a shaving biopsy was performed. He was diagnosed with retinoblastoma of the right eye at 35 months, and the *RB1* gene mutation was not detected. He was treated with ifosfamide, carboplatin, etoposide, vincristine, and actinomycin for the rhabdomyosarcoma. The patient remained off therapy without recurrence for 81 months.

Late cosmetic ophthalmological complications have also been reported. Isolated mild ptosis was recorded in two (10%) patients, anophthalmic enophthalmos in two (10%), and both ptosis and enophthalmos in two (10%).

## 4. Discussion

In this study, chemotherapy with CVDM as a post-enucleation chemotherapy for advanced intraocular retinoblastoma proved to have excellent survival outcomes with tolerable toxicity.

Multiple prospective studies have been conducted on post-enucleation chemotherapy in unilateral high-risk retinoblastoma, which are summarized in Table 3 [13,14,15,16,17]. These all showed excellent outcomes of OS and EFS of 95–100% and 94–100%, respectively. The very few previously described events were mostly recurrences, both locally and systemically, in similar proportions and mostly in the high-risk groups of each study. Our study also showed similar outcomes, with similar indications for adjuvant chemotherapy.

All our patients were alive at the time of analysis and none relapsed after chemotherapy, including the two with unequivocal high-risk criteria since the 1980s, such as scleral invasion and surgical margin invasion, and eight with widely accepted high-risk criteria and concomitant ONI and CI of any degree [18,19]. Chévez-Barrios et al. reported that 3 out of 15 patients with both PLONI and massive CI > 3 mm died despite the administration of intensive chemotherapy, 2 due to CNS recurrences, and 1 due to an unknown cause [13]. This is supposedly because of the high probability of tumor microinvasion of the CSF. The conversely favorable results of the five of our cases with both PLONI and CI may, in part, be attributed to the intensive intravenous chemotherapy and additional intrathecal chemotherapy included in our regimen, although the numbers are small and direct comparison is impossible.

There is a moderate controversy over the specific criteria for determining candidates for adjuvant chemotherapy in retinoblastoma. Chantada et al. were among the first to suggest a graduated intensity approach, describing good prognosis despite withholding adjuvant chemotherapy in patients with isolated PLONI, isolated CI, or isolated AC invasion in a retrospective study [20,21].

Universally accepted low-risk criteria subject to observation are isolated focal CI and isolated prelaminar ONI only. None of our cases fit these criteria, which shows the relatively high disease burden of our group of patients. Only one patient had isolated intra-laminar ONI, which is often regarded as low risk, and previous studies did not report any adverse events in patients with isolated intra-laminar ONI who withheld chemotherapy.

Opinions differ on whether isolated PLONI can be categorized into a separate intermediate-risk group, as candidates for reduced-intensity chemotherapy, or remain in the high-risk group for full intensity therapy. Two studies that adopted the reduced intensity approach and performed four cycles of chemotherapy for those with PLONI as opposed to six cycles for other high-risk features reported a 100% OS and EFS for all risk groups [15,16]. One study found early extraocular relapse in patients with isolated PLONI > 1 mm or >20% of the optic stump and had to reclassify the mentioned factors as high risk [14]. Considering previous results, it is possible that some patients in our high-risk group—for example, some of the nine with isolated PLONI with minimal retrolaminar involvement— did not require such intensive adjuvant chemotherapy. Moreover, patients with isolated intra-laminar invasion may not require chemotherapy.

The literature also shows great controversy regarding the prognosis of isolated massive choroidal invasion and the need for adjuvant therapy; no patients in our study fit this criterion; therefore, no conclusion can be drawn in this aspect [13,17]. The significance of AC invasion is also unclear, because our patient who had AC invasion also exhibited high-risk factors, such as scleral invasion and surgical margin positivity [22].

Regimens for adjuvant chemotherapy in retinoblastoma have varied over the years, and protocols differ worldwide. The agents most often used are a combination of carboplatin, etoposide, and/or vincristine, or a combination of vincristine, idarubicin, and cyclophosphamide [11,13,14,15]. Regarding cycles, six to eight cycles seem to be the norm for high-risk groups, and some propose four or even two cycles for intermediate groups. Intrathecal chemotherapy and ocular radiotherapy are usually reserved only for tumors that extend beyond the sclera or surgical margin [11,23,24]. Radiotherapy is losing popularity due to the high incidence of late endocrinological side effects, as well as the reduced number of advanced diseases with earlier detection.

Our institution has adopted the CVDM regimen since the 1990s, with a prolonged 57-week course of vincristine, doxorubicin, and cyclophosphamide, which has the advantage of an outpatient setting (Figure 2) [12]. The treatment period was longer than that of the other protocols, but the acute toxicities were decreased by reducing the dose of the chemotherapeutics administered at one time was reduced. Routine administration of intrathecal methotrexate is another major difference from other protocols. We hypothesize that this may be effective in controlling the possibility of CNS microinvasion, even when CSF examination is negative for tumors. However, we cannot draw a definite conclusion on the efficacy of intrathecal chemotherapy because it was not verified in a prospective randomized study.

Interestingly, no prospective studies on the follow-up of post-enucleation chemotherapy of unilateral retinoblastoma have reported a case of secondary neoplasm. Secondary malignancy in retinoblastoma has been well described, with an incidence ranging from 8.4% at 18 years post-diagnosis to 90% at 40 years of age. The risk is largely confined to hereditary, usually bilateral, cases with mutant *RB1* due to genetic predisposition [25,26]. However, the development of second non-ocular tumors is partly attributed to treatment, such as external beam radiation before 12 months of age or chemotherapy with alkylating agents, anthracyclines, or topoisomerase inhibitors [27,28]. The most common tumors are osteogenic sarcomas, soft tissue sarcomas, and malignant melanomas [25]. Although rare, cases of chemotherapy-related acute myeloid leukemia have also been reported [29]. Studies unequivocally advocate prompt, aggressive treatment of secondary non-ocular tumors in patients with retinoblastoma. Surprisingly, our patient who developed rhabdomyosarcoma was negative for *RB1* gene mutations, but since this testing was performed in 2006, it may have been an incomplete evaluation yielding a false-negative result. In addition to genetic predisposition, the possibility that the use of alkylating agents, such as cyclophosphamide, contributes to the development of secondary cancer cannot be ignored. Further analysis is warranted to determine the significance and risk factors of secondary malignancy in this patient group as well as various possible late effects.

Limitations of this study include the retrospective nature of the study, the small number of cases, and incomplete long-term follow up data. The small number of cases is largely attributable to the low incidence of advanced retinoblastoma requiring enucleation and chemotherapy. Nowadays, upfront enucleation is uncommon as increasing number of patients are diagnosed in the early stages, and even in advanced cases there are alternative treatment modalities with eye-salvaging potential available. Although patients were followed up for a considerable period of time (median 102.24 months) assessment of endocrinological, neurodevelopmental, psychosocial late effects were mostly not available at the time point of this study. Surveillance for the aforementioned areas is now routinely carried out for newly diagnosed patients.

## 5. Conclusions

In conclusion, our study highlights the effectiveness and safety of the current CVDM regimen for unilateral advanced retinoblastoma, especially in prevention of CNS recurrence. Although treatment period is longer than other regimens commonly used and includes intrathecal chemotherapy, this regimen can be adopted in an outpatient setting and lower doses of chemotherapeutics are administered per visit. Review of treatment trends for post-enucleation retinoblastoma patients warrants further risk stratification within this group and attempts at de-escalation of treatment. A rare case of secondary rhabdomyosarcoma in a unilateral, non-hereditary retinoblastoma patient is described, whose significance is unclear on its own. Continued systematic long-term follow-up is required to further determine any late effects.

## Figures and Tables

**Figure 1 children-09-01983-f001:**
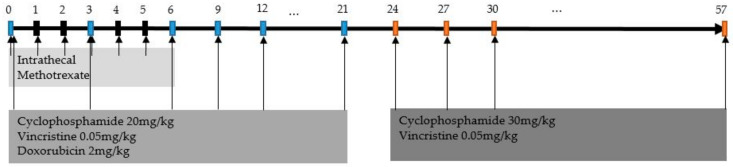
Flowchart of the CVDM regimen.

**Figure 2 children-09-01983-f002:**
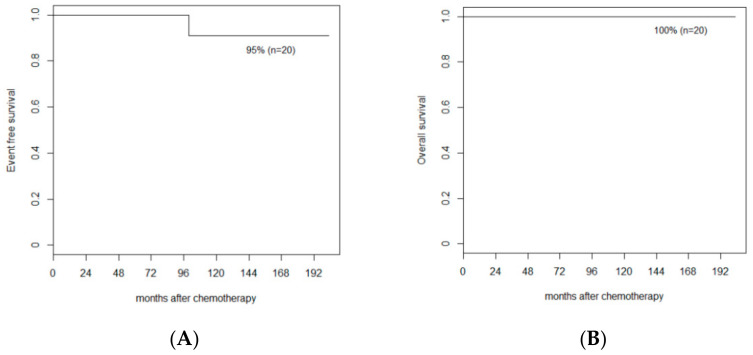
Patient survival. (**A**) Event free survival rate of all the patients. (**B**) Overall survival rate of all the patients.

**Table 1 children-09-01983-t001:** Baseline patient characteristics.

Characteristics	Data (*n* = 20)
Age at diagnosis, Median (range), months	26 (1–45)
Sex, No. (%)	
Male	7 (35)
Female	13 (65)
Laterality, No. (%)	
Right eye	12 (60)
Left eye	8 (40)
Reese–Ellsworth classification, No. (%)	
Group IV	1 (5)
Group V	19 (95)
ICRB, No. (%)	
Group D	2 (10)
Group E	18 (90)
*RB1* mutation analysis, No. (%)	
13q14 deletion	1 (5)
Point mutation	0 (0)
Not detected	11 (55)
Not available	8 (40)
Initial CSF cytology	
No evidence of involvement	20 (100)

Abbreviated: CSF, cerebrospinal fluid; ICRB, intraocular classification of retinoblastoma.

**Table 2 children-09-01983-t002:** Histological high-risk factors (HRFs).

Degree of ONI, No. (%)	Additional High Risk Factors	Data (*n* = 20)
ONI		19 (95)
Pre-laminar		0 (0)
Intra-laminar		3 (15)
	Isolated ONI	1 (33)
	CI	2 (66)
Post-laminar		16 (80)
	Isolated ONI	9 (56)
	CI	5 (31)
	CI and positive surgical margin	1 (6)
	AC and scleral invasion and positive surgical margin	1 (6)
Not available	Uncheckable	1 (5)

Abbreviated: AC, anterior chamber; CI, choroidal invasion; ONI, optic nerve invasion.

**Table 3 children-09-01983-t003:** Summary of prospective studies on post-enucleation chemotherapy of unilateral high-risk retinoblastoma.

Author Year Area	Criteria			Patient No.			Chemotherapy Regimen			Outcome					
	LR	IR	HR	LR	IR	HR	LR	IR	HR	Median FU (Year)	OS (%)	EFS (%)	Recurrence (#)	Second Neoplasm	Death (#)
Chantada et al. 2010 Argentina [14]	CI and/or pre/intra- laminar ONI OR PLONI ^†^	Not defined	Sclera and/or margin+ and/or PLONI + massive CI	65	Not defined	30	Not done	Not defined	4 × CE 4 × VICy RT for margin+	4.1 (0.4–7.6)	LR 96 HR 96	LR 94 HR 96	LR CNS (1) systemic (2) HR CNS (1)	0	LR (2) ^†^ HR (2)
Aerts et al. 2013 France [15]	focal CI and/or pre- laminar ONI	AC and/or massive CI and/or intra-/PLONI	Sclera and/or margin+	70	52	1 *	Not done	2 × CE 2 × VCy	3 × CE + IT thiotepa 3 × VCy CarboPEC^*^	5.9 (2.1–10)	100	100	0	0	0
Sullivan et al. 2014 USA [16]	Focal CI OR pre-/intra- laminar ONI	AC and/or massive CI and/or PLONI and/or ONI + CI	Sclera and/or margin+	36	7	3	Not done	4 × VDC	3 × VDC 3 × VCE	3.4 (0.8–6.4)	100	100	0	0	0
Perez et al. 2018 Latin America [17]	CI and/or pre-/intra- laminar ONI	Not defined	Sclera and/or margin + and/or PLONI	84	Not defined	42	Not done	Not defined	4 × CE 4 × VICy RT for margin+	3.8 (0.4–8.2)	LR 100 HR 95	LR 99 HR 95	LR ocular (1) HR CNS (1)	0	LR (0) HR (2)
Chevez-Barrios et al. 2019 USA [13]	Focal CI OR pre-/intra- laminar ONI	Not defined	Sclera and/or margin + and/or massive CI and/or PLONI and/or ONI + CI	216	Not defined	94	Not done	Not defined	6 × CEV	4	LR 100 HR 97	LR 99 HR 96	LR ocular (1) systemic (1) HR CNS (2) unknown (1)	0	LR (0) HR (3)

AC, anterior chamber; C; carboplatin; CI, choroidal invasion; CNS, central nervous system; Cy, cyclophosphamide; E, etoposide; HR, high risk; I, idarubicin; IR, intermediate risk; IT, intrathecal; LR, low risk; PLONI, postlaminar optic nerve invasion; RT, radiotherapy; V, vincristine. * Only one patient with atypical high-risk histology received CarboPEC and stem cell rescue as consolidation. † Two patients in the low-risk group with isolated PLONI suffered extraocular relapse and death. After interim subgroup analysis, isolated PLONI > 1 mm beyond the lamina cribrosa or >20% of the overall optic stump) was re-assigned to the high-risk group.

## Data Availability

The data presented in this study are available upon request from the corresponding author. The data are not publicly available because of the institutional security policies.

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
