# Peer review of "Twenty-Year Retrospective Study of Post-Enucleation Chemotherapy in High-Risk Patients with Unilateral Retinoblastoma"

_children, 2022, doi:10.3390/children9121983_

Round 1
Reviewer 1 Report
In this 20 year retrospective study, Sunwoo et al, concludes that chemotherapy regime containing cyclophosphamide, vincristine, doxorubicin and methotrexate can have a significant outcome on survival with low toxicity in post-enucleated retinoblastoma patients.
Major weakness of the study is low sample number, n=20 which is considerably low for a retrospective study to make any significant conclusion.
Introduction is poorly written with no mention of relevant studies that have previously used this chemotherapy regime in retinoblastoma.
Please reconsider the title, it’s not very clear. Title should reflect that this study is a 20 year retrospective study.
(D)oxorubicin is a type of (A)nthracycline compound. I suggest authors should consider changing the abbreviation CVAM to CVDM, to avoid the confusion among reader for A.
The exclusion criteria for the study is not mentioned. Flow chart for the treatment regime might be helpful.
In many places authors assumed the readers to have a prior knowledge about the pathogenesis of retinoblastoma, grades of toxicity etc. In result section 3.3; authors should explain about the toxicity grades. Authors cannot expect the readers to be aware of these. Should describe in 2-3 sentences about febrile neutropenia. What does increase in alanine transferase signifies.
Table 1, total no. of patients in RB1 mutation analysis is greater than 20. Please check. Table 3 should be moved to result section.
Conclusion of the study is not very clear. Also, authors have not discussed the limitation of their study.
Author Response
Point 1: Major weakness of the study is low sample number, n=20 which is considerably low for a retrospective study to make any significant conclusion.
Response 1: These 20 cases described are a product of extensive review of electronic medical records spanning 20 years in a large tertiary children’s hospital in Korea. The small number of cases was inevitable due to the very low incidence of advanced retinoblastoma requiring enucleation and chemotherapy. Upfront enucleation is uncommon as increasing number of patients are diagnosed in the early stages, and even in advanced cases there are alternative treatment modalities with eye-salvaging potential.
Point 2: Introduction is poorly written with no mention of relevant studies that have previously used this chemotherapy regime in retinoblastoma.
Response 2: Introduction has been revised to include citation of a previous study with CVDM regimen. Other international studies using similar chemotherapeutics but in a higher dose, inpatient setting are mentioned in detail in the discussion and through table 3.
Point 3: Please reconsider the title, it’s not very clear. Title should reflect that this study is a 20 year retrospective study. (D)oxorubicin is a type of (A)nthracycline compound. I suggest authors should consider changing the abbreviation CVAM to CVDM, to avoid the confusion among reader for A.
Response 3: Changes in title and abbreviations have been made in accordance with reviewer suggestions.
Point 4: The exclusion criteria for the study is not mentioned. Flow chart for the treatment regime might be helpful.
Response 4: I have described the exclusion criteria, drafted and added a treatement flowchart figure.
Point 5: In many places authors assumed the readers to have a prior knowledge about the pathogenesis of retinoblastoma, grades of toxicity etc. In result section 3.3; authors should explain about the toxicity grades. Authors cannot expect the readers to be aware of these. Should describe in 2-3 sentences about febrile neutropenia. What does increase in alanine transferase signifies.
Response 5: I have added additional information on the pathogenesis of retinoblastoma in the introduction and further explanation on toxicities in the result section.
Point 6: Table 1, total no. of patients in RB1 mutation analysis is greater than 20. Please check. Table 3 should be moved to result section.
Response 6: These errors have been addressed and corrected.
Point 7: Conclusion of the study is not very clear. Also, authors have not discussed the limitation of their study.
Response 7: The conclusion has been modified to state more clearly the key advantages of the CVDM regimen, takeaway points from the literature review, and the significance of secondary neoplasm. Limitations of this study such as its retrospective nature, low number of cases, incomplete long term follow up data, have been added to the discussion.
Reviewer 2 Report
1. Line 28 - People in rather than peoples
2. References should be mentioned in the superscript format in the text with the details mentioned at the end.
3. Line 291- Change the word priori to prior
Author Response
Point 1: Line 28 - People in rather than peoples. Line 291- Change the word priori to prior
Response 1: These errors has been corrected.
Point 2: References should be mentioned in the superscript format in the text with the details mentioned at the end.
Response 2: The revised manuscript now has references in superscript format.
Round 2
Reviewer 1 Report
Authors have made a significant effort to improve the manuscript. The discussion and conclusion reads well.
Few minor points to consider:
The flowchart should be Fig.1 as it appears first in the manuscript and patient survival graph should be Fig.2. Accordingly the fig number in line 87 & 139 should be changed.
The font in the flow chart should be the same as the text. Currently it is too small to read clearly.
Author should make the abbreviation CTCAE in parenthesis in line 96, as they have used this abbreviation later in line 146, 3.3-toxicity.
In line 147, it should be surpassing instead of “urpassing”. Authors should recheck for other minor spelling and grammatical check through the manuscript carefully.
Author Response
We thank you for your careful review.
Figure numbers and font in the flowchart have been revised, and abbreviations have been placed in order. This manuscript has been thoroughly screened once more for grammatical and spelling errors.